# Synergistic Control of Transmitter Turnover at Glycinergic Synapses by GlyT1, GlyT2, and ASC-1

**DOI:** 10.3390/ijms23052561

**Published:** 2022-02-25

**Authors:** Volker Eulenburg, Swen Hülsmann

**Affiliations:** 1Department for Anesthesiology and Intensive Care, Faculty of Medicine, University of Leipzig, Liebigstraße 20, D-04103 Leipzig, Germany; 2Department for Anesthesiology, University Medical Center, Georg-August University, Humboldtallee 23, D-37073 Göttingen, Germany

**Keywords:** glycine, transporter, glycinergic inhibition, homeostasis, inhibitory synapse

## Abstract

In addition to being involved in protein biosynthesis and metabolism, the amino acid glycine is the most important inhibitory neurotransmitter in caudal regions of the brain. These functions require a tight regulation of glycine concentration not only in the synaptic cleft, but also in various intracellular and extracellular compartments. This is achieved not only by confining the synthesis and degradation of glycine predominantly to the mitochondria, but also by the action of high-affinity large-capacity glycine transporters that mediate the transport of glycine across the membranes of presynaptic terminals or glial cells surrounding the synapses. Although most cells at glycine-dependent synapses express more than one transporter with high affinity for glycine, their synergistic functional interaction is only poorly understood. In this review, we summarize our current knowledge of the two high-affinity transporters for glycine, the sodium-dependent glycine transporters 1 (GlyT1; SLC6A9) and 2 (GlyT2; SLC6A5) and the alanine–serine–cysteine-1 transporter (Asc-1; SLC7A10).

## 1. Introduction

Glycine (Gly), the simplest amino acid, was first postulated to acts as a neurotransmitter in the central nervous system (CNS) in 1965 [1]. In the mitochondria, glycine can be synthesized by the serine hydroxymethyltransferase (SHMT) that catalyzes the transfer of the hydroxymethyl sidechain from serine to H_4_-folate (THF) to form glycine and 5,10-methylene H_4_-folate [2,3,4,5,6]. At the glycinergic synapses of the mature CNS, glycine is transported into synaptic vesicles by the vesicular inhibitory amino acid transporter (VIAAT or VGAT, SLC32A1) [7]. After exocytotic release into the synaptic cleft, glycine activates postsynaptic strychnine-sensitive glycine receptors (GlyR) leading to the opening of an intrinsic anion channel, which results—at least at most synapses of the mature CNS—in chloride influx and thereby in the hyperpolarization of the postsynaptic membrane and raises the threshold for neuronal excitability [8,9]. High densities of glycinergic synapses can be found in the spinal cord and brain stem, where glycine constitutes the dominant inhibitory neurotransmitter, and in the retina [4,6]. Additionally, some glycinergic synapses are also found in forebrain regions like the hippocampus and the cortex [10,11]. Their functions here, however, are only incompletely understood [4,6].

In addition to its role as an important inhibitory neurotransmitter, glycine also contributes to the regulation of excitatory glutamatergic neurotransmission. Here, glycine binds, alongside D-serine as an essential co-agonist of glutamate, to the N-methyl-D-aspartate receptor (NMDAR) subtype of ionotropic glutamate receptors [12,13,14]. Consequently, it has been shown that most NMDAR cannot be activated without occupancy of this glycine binding site by glycine or by alternative glycine-binding-site agonists like D-serine [15,16].

This complex function of glycine as a neurotransmitter illustrates the importance of a precise regulation of its concentration in the various intracellular and extracellular compartments. In addition to the above-mentioned vesicular VIAAT (or VGAT (SLC31A1)) that catalyzes the import of glycine into synaptic vesicles [7], there numerous plasma membrane transporters that accept glycine as a substrate are known.

Of these transporters, at least three display a high affinity for glycine and are expressed in the vicinity of glycine-dependent synapses. These are the glycine transporters GlyT1 (SLC6A9) and GlyT2 (SLC6A5) and the alanine–serine–cysteine-1 transporter (Asc-1 or SLC7A10, see also Figure 1 and Figure 2). In addition, the neutral amino acid exchanger (ASCT1 or SLC1A4) has been shown by loss-of-function studies to contribute to glycine homeostasis in the brain [17]. Recent studies, however, demonstrated that glycine itself is not a substrate of ASCT1 [18], suggesting that the effect of ASCT1 deficiency is most likely indirect.

In this review, we will first summarize our current understanding of the functions of GlyT1, GlyT2, and ASC1 and subsequently discuss the implications of possible interactions of the individual transporters for CNS function. For the sake of clarity, we will focus on inhibitory synapses.

## 2. Expression Pattern of the Individual Transporters, with a Special Emphasis on the Cell Type Expressing These Transporters

Although some GlyT1 expression can be found in all major brain regions, the highest expression of this transporter is found in astrocytes of the spinal cord and brainstem, i.e., in regions that are rich in inhibitory glycinergic neurotransmission [19,20]. Additionally, GlyT1 has been shown to be expressed in a subpopulation of presumptive glutamatergic neurons of the hippocampus [21]. In vitro data using membrane preparations from caudal brain regions suggested that GlyT1 might also be expressed in glycinergic neurons of the brain stem and spinal cord [22]. These data, however, could not be verified by in vivo approaches [23]. Interestingly, GlyT1 is found in glycinergic amacrine cells of the retina [23,24].

GlyT2 is expressed exclusively in glycinergic neurons and is currently the only reliable marker of glycinergic neurons [25]. It is found almost exclusively in caudal regions of the CNS, thus paralleling the expression pattern of GlyRs [26,27]. Additional GlyT2 immunoreactivity was identified in forebrain regions including the hippocampus and the cortex [10,28]. GlyT2 is localized at the presynaptic terminals juxtaposed to postsynaptic specializations containing GlyRs [28,29]. Moreover, ultrastructural studies demonstrated that the transporter is localized in the pre-synapse surrounding the transmitter release zone juxtaposing postsynaptic specializations containing GlyRs [30] (see Figure 1).

For Asc-1 a neuronal expression was described initially [31]. Indeed, this neuronal role of Asc-1 was supported by different functional studies in the forebrain [32] and brainstem [33]. However, Asc-1 is also strongly expressed in astrocytes [34]. Moreover, *Asc-1* was found among the astrocyte enriched genes [35], and there is a regional heterogeneity of astroglial Asc-1 expression, showing significantly higher Asc-1 levels in the brain regions rich in glycinergic neurotransmission such as the brainstem as compared to other regions such as hippocampus or cortex [36], further supporting the hypothesis that Asc-1 plays an important role in the regulation of glycinergic neurotransmission.

## 3. Structure and Mode of Transport

Both glycine transporters GlyT1 (Slc6a9) and GlyT2 (Slc6a5) belong to a sodium- and chloride-dependent neurotransmitter transporter family, which also include transporters for GABA, catecholamines, serotonin, and proline [37]. They have 12 membrane-spanning domains, connected by six extracellular and five intracellular loops, with intracellularly located C and N termini. The first structure information on these transporters came from X-ray crystallographic studies on the bacterial ortholog leucine transporter LeuTa [38]. These studies revealed that the core of the transporter consists of 10 transmembrane domains that form 2 structurally conserved inverted repeats, with essential parts of substrate binding sites being formed by an unwound region in the transmembrane domains 1 and 6. The validity of the structural models of GlyT1 on the basis of LeuTa was recently demonstrated by X-ray crystallographic data of a truncated human GlyT1 [39].

Although both GlyTs have a high sequence identity and an even higher sequence similarity, GlyT1 and GlyT2 display different transport stoichiometries. While GlyT1 requires 2Na^+^/1Cl^−^ to transport one molecule of glycine, GlyT2 uses 3Na^+^/1Cl^−^ [40], (Figure 2). Based on these differences in stoichiometry and thermodynamic considerations, it was proposed that in contrast to GlyT2, GlyT1 is capable of reverse transport, i.e., it allows a transporter-mediated glycine release that is triggered by changes in ion gradients, membrane potential, or intra- and extracellular glycine concentrations [41]. This hypothesis is supported by studies performed in cultured astrocytes [42]. Up to now, however, evidence for a physiological function of this reverse transport mode of GlyT1 is still missing.

The *Asc-1* (*Slc7a10*) gene encodes the light chain (or catalytic chain) of a hetero(di)meric amino acid transporters (HAT) [43]. Here, the interaction partner, e.g. a heavy chain like 4F2hc/Slc3a2, is essential for the translocation of the transporter to the plasma membrane. The transport function of the Asc-1 transporter appears to be independent of this interaction. Asc-1 has been characterized as a sodium-independent neutral amino acid antiporter (Figure 2), which, in addition to glycine, transports other neutral amino acids such as Alanine, Threonine, Cysteine, and D-serine [44]. To this end, Asc-1 functions as an amino acid exchanger. Initial functional data suggest that the transition from the inside to the outside open conformation (and vice versa) is facilitated by the binding of an amino acid substrate [44]. Thereby, the directionality of (glycine) transport is (energetically) coupled to the concentration gradient of the anti-ported amino acid.

## 4. Genetical and Pharmacological Evidence for the Function of Individual Transporters

### 4.1. GlyT1

Much information on the precise functions of each of these transporters was obtained by loss-of-function approaches. It was demonstrated that GlyT1 contributes, at least in the neonatal animal, to the clearance of glycine from the synaptic cleft [45]. Consistently, the loss of GlyT1 expression in constitutive KO mice resulted in the accumulation of glycine in the extracellular space that thereby facilitated the opening of GlyRs, ultimately leading to the suppression of motor activity including respiration [45] and mimicking major parts of the neurological phenotype seen in animals with nonketotic hyperglycinemia [46]. A similar phenotype was observed in animals lacking only glial GlyT1, demonstrating that the glial expressed GlyT1 controls the extracellular glycine concentration at least in the brain stem and spinal cord of neonatal animals [47], (see Figure 1). Interestingly, phenotypic animals were observed only in the first 7 days after birth, and some animals carrying a glial-specific GlyT1 deficiency survived to adulthood without developing any hypotonia, suggesting that the function of GlyT1 changes during postnatal development [47]. In contrast, neuron-expressed GlyT1 was shown to modulate glutamatergic neurotransmission in various experimental settings [48,49,50], suggesting that the neuronal transporter population is involved in the regulation of the occupancy of the glycine binding site of NMDAR.

In the last couple of years, many highly specific GlyT1 inhibitors have been developed as potential drugs for the treatment of schizophrenia, which was previously associated, on the basis of pharmacological studies, with NMDAR hypofunction [51]. The general idea was that the inhibition of GlyT1 might result in an elevation of the extracellular glycine concentration at excitatory glutamatergic synapses and might thereby facilitate the occupancy of the glycine binding site of NMDARs [52]. Indeed, studies in mice showed that inhibition of GlyT1 resulted in a partial resistance against pharmacologically induced psychosis. A translation of these findings to clinical studies, however, proved difficult, in part due to the large placebo effects observed [53]. In parallel, these substances have been tested in animal models of chronic pain conditions. It was hypothesized that a inhibition of GlyT1 might enhance glycinergic neurotransmission in the dorsal horn of the spinal cord. In this brain region, previous studies demonstrated a loss of synaptic inhibition, and this was suggested to contribute to the chronification of the facilitated pain perception observed in these animal models [54]. Indeed, GlyT1 active substances (both artificial substrates as well as inhibitors) showed anti-hyperalgesic and antiallodynic potential [55,56,57,58,59]. For some substances like the GlyT1 inhibitor Org24598 [60] or the lidocaine metabolite N-Ethyl-glycine [55], it was demonstrated that this effect is at least in part induced in the dorsal horn of the spinal cord, consistent with the hypothesized effect on glycinergic neurotransmission.

Interestingly, there is also neuronal GlyT1 expression in retinal glycinergic amacrine cells [24]. Here, in the absence of GlyT2, GlyT1 expression is sufficient to accumulate glycine intracellularly at concentrations that allow for an efficient loading into vesicles [23]. These findings demonstrate that, at least in this specialized cell type, the 3:1 stoichiometry of GlyT2, that was proposed to be essential for the generation of presynaptic concentrations high enough for efficient vesicle loading, is not required. Whether the 2:1 stoichiometry of GlyT1 would suffice for replenishing glycine in presynaptic terminals in other brain regions is yet unknown.

### 4.2. GlyT2

Mice deficient for GlyT2 appear normal at birth but develop an acute progressing neuromotor disorder, characterized by muscular rigidity, tremor, and hindfeet clasping [61], reminiscent of the phenotype of mice carrying mutations associated with GlyR function impairment [62,63,64]. Consequently, GlyT2^−/−^ mice die in the 2nd postnatal week. Consistent with the phenotype, significant alterations in glycinergic neurotransmission were observed. Recordings of synaptic activity in cultured spinal neurons from GlyT2-KO mice revealed a significant reduction of the amplitude of miniature inhibitory synaptic currents (mIPSCs) but no reduction of their frequency, which is consistent with a reduced filling of the vesicles in GlyT2-deficient mice [61]. Indeed, this conclusion is also supported by the significant reduction of intracellular glycine immunoreactivity [65]. In recordings of synaptic activity in acute brain slices containing the hypoglossal nucleus from WT and GlyT2 KO mice, however, a significant reduction not only of the amplitude but also of the frequency of spontaneous inhibitory synaptic currents (sIPSCs) and miniature postsynaptic currents (mIPSCs) was observed [61]. This discrepancy was attributed to the higher noise level in the recordings in the slice preparations that precluded the detection of small synaptic events. Interestingly, the reduction of the sIPSCs amplitude was already evident immediately after birth, thus preceding the onset of the apparent phenotype [65], suggesting functional changes in the importance of glycinergic neurotransmission during this developmental stage.

The observation that GlyT2 deficiency results in a dramatic reduction in glycine release form the glycinergic presynaptic terminal demonstrates unequivocally that the majority of glycine for presynaptic release is not synthetized by the glycinergic neuron but is taken up from the extracellular space (Figure 1). Thereby, these data provided the first evidence that GlyT2 contributes to a glycine cycle allowing for the recycling of previously released glycine [66].

A recent study suggested that in GlyT2-KO mice, a subset of synaptic vesicles at glycinergic synapses has normal glycine content, at least at the synapse between the medial nucleus of the trapezoid body and the lateral superior olive (MNTB–LSO synapse) [67]. In this case, similar to previous findings [61], both sIPSC amplitude and frequency were significantly reduced. The quantal size q, however, as estimated by gaussian fits of the first peak of the amplitude histogram, was reported not to be changed. Based on these findings, the authors concluded that a subset of glycinergic synaptic vesicles were loaded normally, while others were not loaded at all. This interpretation would require a tight physical and functional coupling of a glycine-synthetizing enzyme (e.g., SHMT, which is localized in the mitochondria) or an alternative uptake transporter (e.g., Asc-1) to individual vesicles. Another explanation, following also the principle of parsimony, is that small-amplitude IPSCs are not detected in the rather “noisy” recording situation in slices. This interpretation is consistent with data from cultured neurons of GlyT2-KO mice, where no reduction of the frequency can be observed [61].

Despite this discrepancy, it is apparent that the loss of GlyT2 does not completely prevent some vesicular loading of glycine [61,65,67]. Possible sources for this residual glycine are locally synthesized glycine by SHMT, but also other transporters that accept glycine as a substrate, such as some transporters of the SNAT family [68] or the ASC-1 transporter [69].

Consistent with the mouse genetical data, pharmacological blockade of GlyT2 in the rat dorsal horn using ORG25543 has been shown to lead to a reduction of the amplitude of glycinergic sIPSC [60]. This, however, requires a long administration of the drug and, e.g., persisting stimulation of the release [60]. In other experimental designs, a similar reduction of IPSC amplitude was not observed [67,70]. In some studies, a tonic inward current was induced by GlyT2 blocker application [60,70], which was absent in recordings from slices of GlyT2^−/−^ mice [61,65]. These events are caused most likely by a transient increase of extracellular glycine concentration, resulting from the reduced uptake into the glycinergic pre-synapse. This is consistent with behavioral data in animal models for chronic pain that demonstrate that a partial pharmacological inhibition of GlyT2 results in the amelioration of hyperalgesia and allodynia, most likely by a local elevation of extracellular glycine concentration at inhibitory glycinergic synapses [71]. This observation supports the concept that in the mature nervous system, GlyT2 contributes to the concentration of glycine in the extracellular or synaptic space. In this context, it is noteworthy that a complete inhibition of GlyT2 results in severe side effects such as respiratory disturbances and plegia [72], consistent with a breakdown of glycinergic neurotransmission, thus confirming the essential role of GlyT2 for presynaptic transmitter replenishment [61].

Interestingly, acute pharmacological blockade of GlyT2 by ALX1393 (2 µM) exerted a small effect on the time course of synaptic depression during high-frequency stimulation in the lateral superior olive (LSO; Brill et al. 2021). While this observation could be explained by the existence of a large reserve pool of glycine-pre-filled vesicles in LSO glycinergic neurons, the significant slowing of recovery from depression, however, underlines the acute effect of GyT2 blockade on providing substrates for VIAAT-dependent vesical refilling [67].

The existing data suggest that, although GlyT2 also contributes (to some extent) to the regulation of extracellular glycine concentration at inhibitory synapses, its main function is the replenishment of presynaptic glycine for vesicle refilling. In this context, a chronic depletion of presynaptic glycine results in a reduced vesicular glycine release from the glycinergic pre-synapse. Thereby, the complete and long-lasting loss of GlyT2 function in KO mice is predicted to cause reduced extracellular glycine levels rather than elevated ones. The precise contribution of GlyT2 to the control of extracellular glycine concentration at glycinergic synapses, however, is currently not clear and deserves further investigation.

### 4.3. Asc-1

Similar to the phenotype seen in GlyT2^-/-^ mice, Asc-1 KO mice show a motor dysfunction that resembles human hyperekplexia [69]. Indeed, glycinergic synaptic transmission was found to be reduced in the hypoglossal nucleus [69] and in the spinal cord [34]. Moreover, Asc-1 KO mice display reduced levels of glycine in the brain [69]. The results of these studies are consistent with the idea that the loss of the Asc-1 leads to a reduction in the vesicular filling of synaptic vesicles. Taken together, these findings led to the conclusion that Asc-1 is, like GlyT2, an essential part of the glycine shuttle system and is thus essential for the import of glycine into glycinergic pre-synapses in vivo [69].

However, acute blockade of Asc-1 with a non-transportable antagonist (Lu AE00527) did not impair glycinergic transmission [33], whereas application of a transportable/agonist antagonist (D-isoleucine) led to the fast depletion of the neuronal presynaptic glycine pool and consequently to a dramatic reduction of glycinergic synaptic transmission in brainstem slices [33]. In contrast to the inhibition of GlyT1 or GlyT2, no significant change in the holding current was described after perfusion with Asc-1 blockers or Asc-1 substrates such as D-Ile [33]. The discrepancy of the effects of D-isoleucine and Lu AE00527 demonstrates that at least under resting conditions, neuronal Asc-1 does not contribute to the uptake of glycine into the pre-synapse and thereby to the refilling of synaptic vesicles with glycine. The relatively fast onset of the D-ILE effect, however, could be explained by an Asc-1-dependent release of glycine from the presynaptic terminal. Whether this causes a secondary release of glycine, packed already in the synaptic vesicles into the presynaptic terminal via reversed VIAAT transport, or whether the fast rundown of synaptic current is caused by a very fast turnover rate of the synaptic vesicles at hypoglossal glycinergic synapses is unclear at present.

Taken together, these findings demonstrate that the function of Asc-1 expressed by glycinergic neurons is not connected directly to the glycine cycle at glycinergic synapses. In the presynaptic neuron, Asc-1 might use the steep glycine gradient across the membrane as a driving force for the uptake of other amino acids for, e.g., local protein synthesis [73], thus functioning as a glycine-driven amino acid transporter (Figure 2). Similarly, Asc-1 might contribute to the uptake of L-Cysteine for glutathione synthesis and thus to the control of oxidative stress in the presynaptic terminal [74].

Despite its unclear functions at glycinergic neurons [34], especially the characterization of Asc-1-deficient mice clearly demonstrates that Asc-1 has a previously unrecognized essential role in glycine turnover at glycinergic synapses. Although direct experimental evidence is still lacking, the current data are consistent with the idea that Asc-1 mediates the release of glycine from astrocytes to provide a substrate for GlyT2-mediated glycine uptake into the glycinergic pre-synapse (Figure 1). In the Asc-1 KO mouse characterized by long-term Asc-1 deficiency [69], the observed loss of tissue glycine might be explained by the inability to release glycine from astrocytes to provide a substrate for presynaptic GlyT2-mediated uptake. This might subsequently induce astroglial degradation by the glycine cleavage system [75] or release into the bloodstream via other low-affinity transporters. In conflict with these hypothesis, an acute pharmacological block of astroglial Asc-1 did not result in the expected reduced refilling of the glycinergic vesicles in neighboring neurons [33]. One possible explanation is the rather short incubation time (20 min) before measuring the spontaneous IPSC. Taken together, both mouse genetical and pharmacological data clearly demonstrate that Asc-1 has a role in glycine turnover at inhibitory glycinergic synapses that is distinct from that of GlyT1 and GlyT2. Regarding its neuronal function, it appears to us that it is by unlikely that Asc-1 acts primarily as an uptake transporter for glycine as put forward in Safory et al., 2015, and Brill et al., 2021.

## 5. Regulation of the Transport Activity

The functions of the respective transporters suggest that their activity at glycinergic synapses is tightly controlled and might thereby constitute an additional level of regulation of synaptic efficacy. Despite intense research, however, relatively little is known about potential regulation mechanisms. It is assumed that the primary regulation of the total transport activity is achieved by controlling the number of the respective transporters in the plasma membrane. Both stability in the plasma membrane and trafficking of the transporter have been shown to be influenced by direct interactions especially involving the intracellular N- and C-termini of the transporter. Regions of the C-terminus of both GlyT1 and GlyT2 have been shown to be essential for the export of the transporter from the ER [76]. Additionally, protein interactions via the PDZ domain binding motif located in the extreme C-terminus of both transporters have been suggested to contribute to the subcellular targeting of the respective transporter [77,78].

An additional level of regulation of the transporters appears to involve the phosphorylation of the transporter. To this end, it was demonstrated that the surface expression of GlyT1 can be influenced by a number of different kinases including Protein Kinase B (PKB) [79] and Glycogen synthase kinase 3β [80]. For GlyT1, it was described that PKC can alternatively induce its ubiquitinylation and subsequent degradation [81]. Similarly, it was described that GlyT2 ubiquitinylation induced by the signaling molecule Hedgehog is an important mechanism to induce internalization and subsequent degradation [82].

Newer studies suggest that GlyT1 is also downregulated in cultured astrocytes by the activation of the protein kinase AMPK in a most likely BDNF-dependent pathway [83]. In addition to these signaling pathways that regulate the expression of GlyT1 in the plasma membrane, it was shown that there is also a microRNA- (miR) dependent mechanism that regulates the translation of GlyT1 mRNA and thereby, indirectly, the number of GlyT1 molecules within the plasma membrane [84]. The importance of these mechanisms for the in vivo activity of GlyT1 in different cell types, however, still remains to be determined.

In contrast to GlyT1 that appears to be localized predominantly in the plasma membrane, a significant proportion of GlyT2 has been found in intracellular vesicles [85,86], and transporters stored in these vesicles have been shown to be incorporated into the plasma membrane in a Ca^2+^-dependent fashion [85]. Interestingly, Ca^2+^ signals elicited by the activation of the muscarinic acetylcholine receptor M2 result in a reduction of GlyT2 transport activity, most likely due to the internalization of the transporter [87]. If and how an active regulation of GlyT2 contributes to the fine tuning of glycinergic neurotransmission, however, is not clear at present.

It was also postulated that there is an additional regulation of the activity of Asc-1. Up to now, however, relatively little is known about potential mechanisms. Although several potential phosphorylation sites have been identified inside Asc-1 [31], their functional relevance remains, at least up to now, enigmatic.

## 6. Pathological Correlates in Humans

As observed for glycine receptors, also the functions of glycine specific transporter appear to be conserved between rodents and humans. Human genetical data demonstrated that defects of GlyT but also of Asc-1 functionality lead to severe clinical symptoms. Homozygous mutations in the GlyT1 (*SLC6A9*) gene have been shown to cause a disease phenotype very similar but not identical to glycine encephalopathy or nonketotic hyperglycinemia [88,89,90], a disease previously associated with defects in the glycine cleavage system. Similar to patients suffering from glycine encephalopathy, affected individuals have severe neurological problems reminiscent of the phenotype seen in GlyT1-deficient mice, characterized by hypotonia and severe respiratory depression directly after birth. In contrast to glycine encephalopathy patients who show very high glycine concentrations in both the serum and the CSF, patients carrying homozygous function-impairing GlyT1 mutations do not show elevated serum glycine levels, but only a slightly elevated CSF glycine concentration. These findings suggest that in addition to GlyT1, there are additional factors that contribute to the control of extracellular glycine concentration. Whether these factors include GlyT2 and/or Asc-1 or other low-affinity transporters in glia, neurons, or endothelial cells is unclear at present. Based on the differences between classical nonketotic hyperglycinemia patients and those suffering from homozygous function-impairing GlyT1 mutations, the disease resulting from the latter was termed GlyT1 encephalopathy [90]. Interestingly, all GlyT1 encephalopathy patients identified up to now showed, in addition to marked life-threatening neurological disturbances, malformations like clubfeet and arthrogryposis, i.e., phenotypes that have previously not been seen in the respective mouse models. Whether these differences reflect developmental differences or are a result of differences in GlyT1 function between mice and men is unclear at present.

For GlyT2, it was shown that homozygous (or compound heterozygous) nonsense, frameshift, and function-impairing missense mutations of the GlyT2 gene result in hyperekplexia [91,92,93], thus mimicking the consequences of a loss-of-function mutation in the glycine receptor genes GLRA1 or GLRB [94]. This congenital neurological disorder results in the development of hypertonia and exaggerated startle response to tactile or acoustic stimuli within the first postnatal year, that can result in potentially life-threatening episodes of breath-holding. Interestingly, not only homozygous mutations but also some specific heterozygous mutations were shown to result in hyperekplexia, demonstrating the possibility of dominant negative mutations [95]. Recently, the first Asc-1 mutations have been found in hyperekplexia patients [96]. In contrast to homozygous GlyT2-deficient mice that die in the second postnatal week, in adult human hyperekplexia patients, only a relatively mild hyperreflexia persists that in most cases does not need therapeutic intervention [94]; this has been observed for both GlyR and GlyT2 mutations. These differences might result from variations in systems that can compensate for the loss of glycinergic inhibition, such as, e.g., in the GABAergic system.

## 7. Summary

In summary, the data currently available demonstrate that at glycinergic synapses, the neurotransmitter glycine is reused by a neurotransmitter shuttle system that involves both neurons and surrounding glial cells. Here, glycine that is released by a glycinergic neuron, is removed from the synaptic cleft by binding to astrocytic GlyT1, which functions as a buffer system for extracellular glycine. At least a fraction of this bound glycine is then taken up into astrocytes, leading to a locally high intracellular concentration of free glycine. By distribution within the astrocytes, glycine could be released form astrocytes to the blood stream or eliminated by the GCS. Both options, however, would result in a depletion of glycine from the brain. Alternatively, glycine could be released to the extracellular space where it can be taken up by GlyT2. In astrocytes surrounding glycinergic synapses, considerable expression of GlyT1 and Asc-1 is observed [34], suggesting their coexpression within individual astrocytes. Thus, it is possible that glycine that has been taken up via GlyT1 is released from astrocytes via Asc-1 [34], (see Figure 1). In contrast to a release via reverse transport by GlyT1 [97], which has not yet been confirmed under physiological conditions, a redistribution using Asc-1 would not require changes of the sodium gradient. Since Asc-1 functions as an amino acid exchanger, however, a counter-transported amino acid that is taken up from the extracellular space into astrocytes needs to be present, to allow for Asc-1-mediated glycine release.

## 8. Outlook

Although this model of a neuron–glial glycine cycle is very intriguing, many aspects of the precise turnover of glycine at the glycinergic synapse remain speculative. Studies that address if and how Asc-1 is involved in glycine release from astrocytes and the precise functions of neuronal Asc-1 are urgently required. These studies will pave the way to determine functional interactions between the above-discussed glycine-specific transporters and will thereby extend our understanding of glycine turnover at glycinergic synapses.

## Figures and Tables

**Figure 1 ijms-23-02561-f001:**
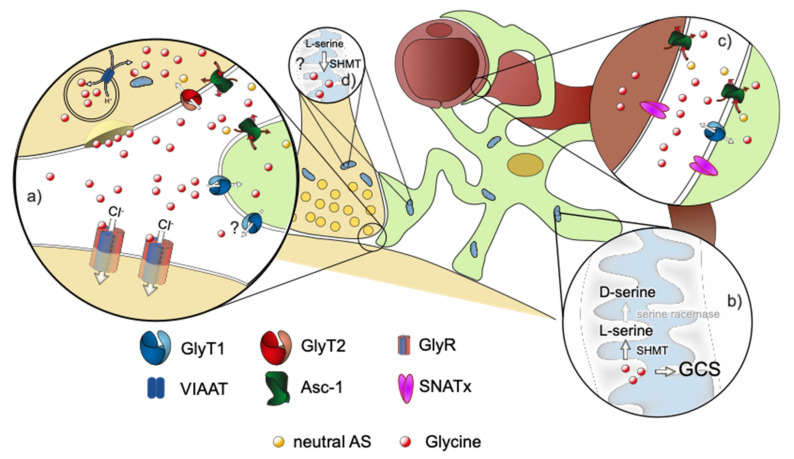
Schematic drawing of a glycinergic synapse. Neuron in beige, astrocyte in green, blood vessel in brown. (**a**) Magnification of the tripartite synapse. Glycine, which is released by the presynaptic neuron activates postsynaptic glycine receptors. It is removed from the synaptic cleft by binding to the astroglial GlyT1, followed by uptake into the astrocyte. Consecutively, glycine can be released from the astrocyte via Asc-1 or possibly via the reverse transport of GlyT1 to the extracellular space, where glycine is transported into the pre-synapse by GlyT2. Excessive glycine can be (**b**) degraded in the astrocytic mitochondrial Glycine cleavage system (GCS) or converted to D-serine by SHMT/SR. Alternatively, glycine can be exported across the blood brain barrier (BBB) via transporters like GlyT1, Asc-1, or members of the SNAT family (**c**). The participating transporters in the process, however, are not known at present. Additional sources of glycine in the CNS are its de novo synthesis in the mitochondria via SMHT using serine as a substrate (**d**) and its import across the BBB (**c**).

**Figure 2 ijms-23-02561-f002:**
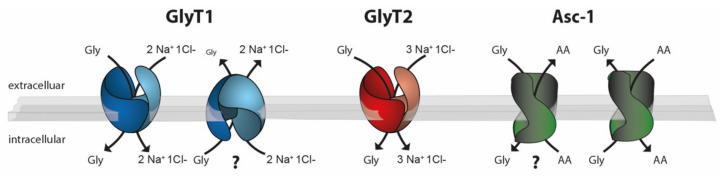
Schematic drawing of the stoichiometries of the glycine transporters GlyT1 (blue), GlyT2 (red), and Asc-1 (green). Whereas GlyT1 and GlyT2 use the Na^+^ Cl^−^ gradient as an energy source for glycine transport, Asc-1-dependent glycine transport is energetically coupled to the counter transport of a neutral amino acid. Experimental evidence suggests that both GlyT1 and Asc-1 can function as glycine importers or exporters. In vivo data demonstrating the physiological relevance of GlyT1-mediated glycine export or Asc-1-dependent glycine import, however, are still lacking (?).

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
