# Peer review of "Synergistic Control of Transmitter Turnover at Glycinergic Synapses by GlyT1, GlyT2, and ASC-1"

_ijms, 2022, doi:10.3390/ijms23052561_

Round 1

Reviewer 1 Report

The review article by Eulenburg and Hülsmann addresses an important topic focussing especially on the high affinity transporters known to be closely involved in regulation of glycinergic inhibitory neurotransmission in the Central Nervous System. In this view, the authors give a picture of possible functional interactions between the three types of transporters named GlyT1, GlyT2 and Asc-1, especially highlighting some characteristics of GlyT2 and, more so, Asc-1 and possible functional interactions. The manuscript helps to fill some gaps present in published review articles on the field; information provided (including the Figure 1) is well presented and seems clear for the reader. Overall this work gives an original contribution to the knowledge on this topic, that should be useful for other scientists and stimulate novel research.

General and specific comments:

The manuscript in part addresses already well known items, in addition the number of cited studies that can be really considered as recent (for example, those published since 2015) appears limited if compared to the total number of studies cited, however most of the recent available research are appropriately mentioned and therefore the quality of the manuscript is good. A few further very recent papers might be mentioned if possible.

Chapter 2:

Lines 79 to 91 : The authors state correctly that GlyT2 is “almost exclusively” found in caudal regions, still a very short note might be added indicating the existence of some “exceptions” to this generally valid statement (see, just for an example, Aroeira et al, 2016, Neurochem. Int. doi: 10.1016/j.neuint.2016.06.007). At the same time, (Lines 87-88) the existence of some relevant, more recent studies indicating the presence of neuronal GlyT1 in mainly glutamatergic neurons (in addition to reference 25) could be briefly acknowledged at this point.

Chapter 2, line 88 . The terms “rather artificial” seem inappropriate, since most “in vitro” techniques are , at least to a certain extent, artificial. It is suggested to remove these terms.

Minor points:

There are some typos :

Introduction, Line 27-28 ; Line 38 “Their”;

Chapter 2, line 88 “In”

Chapter 4, line 151 “is involved” ; Line 214 “exerted”; Line 220 “to some extent”; Line 256: remove “of ”; Line 262 “essential role IN glycine turnover”

Author Response

The review article by Eulenburg and Hülsmann addresses an important topic focussing especially on the high affinity transporters known to be closely involved in regulation of glycinergic inhibitory neurotransmission in the Central Nervous System. In this view, the authors give a picture of possible functional interactions between the three types of transporters named GlyT1, GlyT2 and Asc-1, especially highlighting some characteristics of GlyT2 and, more so, Asc-1 and possible functional interactions. The manuscript helps to fill some gaps present in published review articles on the field; information provided (including the Figure 1) is well presented and seems clear for the reader. Overall this work gives an original contribution to the knowledge on this topic, that should be useful for other scientists and stimulate novel research.

Reply: Thank you for this supporting summary

The manuscript in part addresses already well known items, in addition the number of cited studies that can be really considered as recent (for example, those published since 2015) appears limited if compared to the total number of studies cited, however most of the recent available research are appropriately mentioned and therefore the quality of the manuscript is good. A few further very recent papers might be mentioned if possible.

Reply: We have revised the manuscript and added more recent references.

Chapter 2:

Lines 79 to 91 : The authors state correctly that GlyT2 is “almost exclusively” found in caudal regions, still a very short note might be added indicating the existence of some “exceptions” to this generally valid statement (see, just for an example, Aroeira et al, 2016, Neurochem. Int. doi: 10.1016/j.neuint.2016.06.007). At the same time, (Lines 87-88) the existence of some relevant, more recent studies indicating the presence of neuronal GlyT1 in mainly glutamatergic neurons (in addition to reference 25) could be briefly acknowledged at this point.

Reply: Thank you very much for this suggestion. We have now discussed the mentioned papers in our revision, and incorporated additional references that describe neuronal expression of GlyT1.

Chapter 2, line 88 . The terms “rather artificial” seem inappropriate, since most “in vitro” techniques are , at least to a certain extent, artificial. It is suggested to remove these terms.

Reply: this part was revised

Minor points:

There are some typos :

Introduction, Line 27-28 ; Line 38 “Their”;

Chapter 2, line 88 “In”

Chapter 4, line 151 “is involved” ; Line 214 “exerted”; Line 220 “to some extent”; Line 256: remove “of ”; Line 262 “essential role IN glycine turnover”

Reply: all mentioned typos were corrected!

Reviewer 2 Report

This review summarizes the knowledge regarding different glycine transporters and their interactions during glycine turnover in the CNS.

In general, I think the idea of this review is really interesting and the authors’ observations on this timely topic may be of interest to the readers of the International Journal of Molecular Sciences. However, some comments and more evidence should be included to support the authors’ argumentation. This would improve the quality of the manuscript, its adequacy, and thus its readability prior to the publication in the present form.

General comments

Formatting needs a profound revision as some parts of the text are in a different format as well as some references (e.g. lines 122 – 134). Similarly, English also needs a revision, some parts are hard to follow.

The references used are not as current as IJMS guidelines required. Less than 25% of the references are from 2016 or ahead of (17 out 72 references). It is hard to believe that no more recent studies have been performed regarding glycine transporters during these last years. I urge authors to update their references, even if this requires including more information in the different sections.

I would ask the authors to include a section discussing cellular mechanisms underlying the regulation of the transports; because this field has not assessed, and changes in transporter dynamics as well as its influence of glycine turnover will increase the relevance and quality of this review article.

Figures

The Figure 1 is only cited once, when many parts of the text are referring to it. Include more references of figure 1 according to those parts.

Including a second figure for the section 3 (Structure and mode of transport) would be helpful to visualize all the information from the text.

  1. Introduction

Line 26. Include where the glycine is synthesized.

  1. Expression pattern of the individual transporters with a special emphasis on the cell type which express these transporters

I would recommend to keep the same order when explaining the different transporters. For example, here, authors started talking about GlyT2, then GlyT1 and finally Asc-1. But in other subsequently sections, authors changed and started with GlyT1, then GlyT2 and Asc-1. Since it appears that GlyT1 is the main transporter during embryonic state, being changed by GlyT2 after birth; a general order could be to discuss GlyT1 first, and then GlyT2; as authors did in section 4.

Lines 84 - 92. Overall this paragraph is a bit confusing, I would recommend to rewrite it. For example, the sentence GlyT1 has been shown to be expressed in a subpopulation of … it does not state where these neurons are located.

  1. Structure and mode of transport

Lines 116 - 119. The sentence … it was proposed that in contrast to GlyT2, GlyT1 is capable of … needs a reference at the end. Moreover, what did they see to claim this? How do these differences in stoichiometry mediate the reverse transport? Clarify this.

  1. Genetical and pharmacological evidence for the function of individual transporters

4.1 GlyT1

Again, I would recommend to rewrite it and make this paragraph clearer; some parts are confusing.

Lines 138 - 139. The sentence Here it was demonstrated that GlyT1 contributes at least… needs a reference at the end.

Lines 140 - 143. The sentence Consistently, loss of GlyT1 expression resulted in an accumulation … does it refer to neuronal expression? It is not clearly stated as it does in the next sentence: a similar phenotype was observed in animals lacking only glial GlyT1 … Please, clarify this.

Lines 149 - 152. This last sentence is confusing, rewriting is needed.

Lines 155 - 158. Why cannot the 3:1 stoichiometry be used to explain the evolution of GlyT2? It is not clear.

It is striking authors did not cite/discuss studies regarding drug treatments as they did for GlyT2 and Asc-1. No studies using GlyT1-targeting drugs have been performed so far?

4.2 GlyT2

Define the abbreviature MNTB-LSO synapse, it was not previously done (line 186).

Similar to previous sections, rewriting the text will help to a better understanding. For example, the sentence The observation that GlyT2 deficiency results in a dramatic reduction … (lines 179 – 182) is confusing and contradictory with claims/information made in subsequent sentences (e.g. lines 198 – 201). Furthermore, how can be possible that the presynaptic glycine present in glycinergic neurons is not partly coming from endogenous synthesis? Then, for what reason do glycinergic neurons synthetize glycine?

Moreover, it is firstly claimed that pharmacological inhibition of GlyT2 ameliorates hyperalgesia and allodynia by elevating extracellular glycine levels (lines 211 – 213), but this is in opposition to a following sentence claiming … the complete and long-lasting loss of GlyT2 function in KO-mice is predicted to cause reduced extracellular glycine levels … (lines 224 – 226).  How can this be possible?

Also, how does a inhibition of glycinergic signaling decrease hyperalgesia and allodynia? In this way, would it be stimulating excitatory signaling? How can GlyT2 be mainly implicated in vesicle refilling if it is a plasmatic membrane protein?

4.3 Asc-1

Lines 260 - 265. Not clear enough, and some cites are lacked.

  1. Conclusion

Here authors put together and connect all the previous information, showing the synergy between glycine transporters.  It does not look like a conclusion section, and so, I would recommend to change the title according to the text (GlyTs synergy or something similar); and finally complete the review with an additional section with future studies and concluding remarks.

Author Response

This review summarizes the knowledge regarding different glycine transporters and their interactions during glycine turnover in the CNS.

In general, I think the idea of this review is really interesting and the authors’ observations on this timely topic may be of interest to the readers of the International Journal of Molecular Sciences. However, some comments and more evidence should be included to support the authors’ argumentation. This would improve the quality of the manuscript, its adequacy, and thus its readability prior to the publication in the present form.

General comments

Formatting needs a profound revision as some parts of the text are in a different format as well as some references (e.g. lines 122 – 134). Similarly, English also needs a revision, some parts are hard to follow.

We have to apologize for the formatting of the first version of the manuscript. Apparently, some formatting errors have occurred when pasting the draft in the IJMS format. This has now been corrected.  

The references used are not as current as IJMS guidelines required. Less than 25% of the references are from 2016 or ahead of (17 out 72 references). It is hard to believe that no more recent studies have been performed regarding glycine transporters during these last years. I urge authors to update their references, even if this requires including more information in the different sections.

I would ask the authors to include a section discussing cellular mechanisms underlying the regulation of the transports; because this field has not assessed, and changes in transporter dynamics as well as its influence of glycine turnover will increase the relevance and quality of this review article.

Reply: Thank you very much, we have added a section discussing the current knowledge of transporter regulation. We agree with the reviewer, that in this review many rather old studies are cited. The reason for this is, that we wanted to give credit to the original studies first describing the respective phenomenon and this includes the citing of some even historic studies. The changes suggested by the reviewer, however, has now led to the incorporation of many newer studies, so that we are now confident that the IJMS guidelines are followed.

Figures

The Figure 1 is only cited once, when many parts of the text are referring to it. Include more references of figure 1 according to those parts.

Including a second figure for the section 3 (Structure and mode of transport) would be helpful to visualize all the information from the text.

Reply: We thank the reviewer for this suggestion. We now include a new figure 2 that shows the mode of transport for GlyT1, then GlyT2 and Asc-1 we however refrained from incorporating structural and since we want to keep the focus on the physiology.

Introduction

Line 26. Include where the glycine is synthesized.

Reply: Thank you! This information has been added!

  1. Expression pattern of the individual transporters with a special emphasis on the cell type which express these transporters

I would recommend to keep the same order when explaining the different transporters. For example, here, authors started talking about GlyT2, then GlyT1 and finally Asc-1. But in other subsequently sections, authors changed and started with GlyT1, then GlyT2 and Asc-1. Since it appears that GlyT1 is the main transporter during embryonic state, being changed by GlyT2 after birth; a general order could be to discuss GlyT1 first, and then GlyT2; as authors did in section 4.

Reply: We agree with the reviewer, that keeping the same order enhances the readability of the review. We have therefore adapted the order throughout the manuscript

Lines 84 - 92. Overall this paragraph is a bit confusing, I would recommend to rewrite it. For example, the sentence GlyT1 has been shown to be expressed in a subpopulation of … it does not state where these neurons are located.

Reply: This information is now provided

  1. Structure and mode of transport

Lines 116 - 119. The sentence … it was proposed that in contrast to GlyT2, GlyT1 is capable of … needs a reference at the end. Moreover, what did they see to claim this? How do these differences in stoichiometry mediate the reverse transport? Clarify this!

Reply: this section was rewritten

  1. Genetical and pharmacological evidence for the function of individual transporters

4.1 GlyT1

Again, I would recommend to rewrite it and make this paragraph clearer; some parts are confusing.

Lines 138 - 139. The sentence Here it was demonstrated that GlyT1 contributes at least… needs a reference at the end.

Lines 140 - 143. The sentence Consistently, loss of GlyT1 expression resulted in an accumulation … does it refer to neuronal expression? It is not clearly stated as it does in the next sentence: a similar phenotype was observed in animals lacking only glial GlyT1 … Please, clarify this.

Lines 149 - 152. This last sentence is confusing, rewriting is needed.

Lines 155 - 158. Why cannot the 3:1 stoichiometry be used to explain the evolution of GlyT2? It is not clear.

Reply: We thank the reviewer for these comments, we have rephrased the respective sections. The “evolutionary point” was too far reaching, indeed. We have rewritten the respective sentence. 

It is striking authors did not cite/discuss studies regarding drug treatments as they did for GlyT2 and Asc-1. No studies using GlyT1-targeting drugs have been performed so far?

Reply: Indeed, there have been many studies describing the development and testing of GlyT1 inhibitors. We have not incorporated these studies initially since they predominantly focused on glycine dependent glutamatergic neurotransmission. Thereby, some studies using GlyT1 inhibitors to modulate glycinergic neurotransmission, including some of our own have not been cited. To address this criticism, we have now incorporated an extended paragraph on GlyT1 pharmacology.

4.2 GlyT2

Define the abbreviature MNTB-LSO synapse, it was not previously done (line 186).

Reply: the abbreviation is explained in the revision

Similar to previous sections, rewriting the text will help to a better understanding. For example, the sentence The observation that GlyT2 deficiency results in a dramatic reduction … (lines 179 – 182) is confusing and contradictory with claims/information made in subsequent sentences (e.g. lines 198 – 201). Furthermore, how can be possible that the presynaptic glycine present in glycinergic neurons is not partly coming from endogenous synthesis? Then, for what reason do glycinergic neurons synthetize glycine?

We have to apologize for not being clearer on this point: The reviewer is right. We do not exclude that endogenous glycine synthetized by the  glycinergic neuron is used for vesicular release. However, the available data clearly demonstrate that the synthesis rate of these neurons is by far not sufficient to provide enough glycine to maintain normal glycinergic neurotransmission.

Moreover, it is firstly claimed that pharmacological inhibition of GlyT2 ameliorates hyperalgesia and allodynia by elevating extracellular glycine levels (lines 211 – 213), but this is in opposition to a following sentence claiming … the complete and long-lasting loss of GlyT2 function in KO-mice is predicted to cause reduced extracellular glycine levels … (lines 224 – 226).  How can this be possible?

Also, how does a inhibition of glycinergic signaling decrease hyperalgesia and allodynia? In this way, would it be stimulating excitatory signaling?

Reply: We have to apologize, that our writing was not clear at this point. GlyT2 is essential for the replenishement of the presynaptic vesicle pool but also contributes to the regulation of the extracellular glycine concentration.

We have revised that section, to make this dual function clearer, and to discriminate between the two apparently opposing effects of partial vs. complete GlyT2 inhibition.

How can GlyT2 be mainly implicated in vesicle refilling if it is a plasmatic membrane protein?

Reply: Thank you very much for bringing up this point. At this point our wording has been imprecise. We have now reworded the respective sentence and made it clear, that GlyT2 contributes to the presynaptic transmitter replenishment and thereby provides substrate for the consecutive vesicular filling

4.3 Asc-1

Lines 260 - 265. Not clear enough, and some cites are lacked.

Reply: We have revised that section, to make this clear.

  1. Conclusion

Here authors put together and connect all the previous information, showing the synergy between glycine transporters.  It does not look like a conclusion section, and so, I would recommend to change the title according to the text (GlyTs synergy or something similar); and finally complete the review with an additional section with future studies and concluding remarks.

Thank you very much! We have revised that last para accordingly.

Reviewer 3 Report

This manuscript summarizes the current knowledge about two glycine transporter and the alanine-serine-cysteine-1 transporter (that play a role in synaptic availability of glycine). The authors describe the role of these transporters at glycinergic synapses, their structure and function. At the end of the manuscript, the authors highlight the importance of glycine specific transporters by summarizing the relationship between these transporters and human pathologies. In conclusion, this work contributes to our understanding of glycinergic synapses.

The only modifications I suggest are formatting:

  • Note that there is a small formatting error between lines 122-134 of the manuscript. The font size is different.
  • In the figure there is a spot/line in the lower left corner.
  • In the figure, “b” describes sources of glycine in the CNS by the de-novo synthesis in mitochondria via the SMHT using serine as a substrate and, another “b” shows the conversion of glycine to D-serine by the SHMT/SR in the mitochondria of astrocytes. It would be clearer if one was named as "b" and the other as "d"

Author Response

This manuscript summarizes the current knowledge about two glycine transporter and the alanine-serine-cysteine-1 transporter (that play a role in synaptic availability of glycine). The authors describe the role of these transporters at glycinergic synapses, their structure and function. At the end of the manuscript, the authors highlight the importance of glycine specific transporters by summarizing the relationship between these transporters and human pathologies. In conclusion, this work contributes to our understanding of glycinergic synapses.

Reply: Thank you for this supporting summary

The only modifications I suggest are formatting:

  • Note that there is a small formatting error between lines 122-134 of the manuscript. The font size is different.

Reply: Was changed as suggested

  • In the figure there is a spot/line in the lower left corner.

Reply: The figure has been corrected

  • In the figure, “b” describes sources of glycine in the CNS by the de-novo synthesis in mitochondria via the SMHT using serine as a substrate and, another “b” shows the conversion of glycine to D-serine by the SHMT/SR in the mitochondria of astrocytes. It would be clearer if one was named as "b" and the other as "d"

Reply: Was changed as suggested

Round 2

Reviewer 2 Report

Thanks for addressing all my suggestions/comments.